# Parallel-Probe: Towards Efficient Parallel Thinking via 2D Probing

**Tong Zheng** [1] [*] [†]  **Chengsong Huang** [2] [*] [†]  **Runpeng Dai** [3] [*] [†]  **Yun He** [1] [†]  **Rui Liu** [1]  **Xin Ni**  **Huiwen Bao**
**Kaishen Wang** [1]  **Hongtu Zhu** [3]  **Jiaxin Huang** [2]  **Furong Huang** [1]  **Heng Huang** [1]

## Abstract

Parallel thinking has emerged as a promising paradigm for reasoning, yet it imposes significant computational burdens. Existing efficiency methods primarily rely on local, per-trajectory signals and lack principled mechanisms to exploit global dynamics across parallel branches. We introduce 2D probing, an interface that exposes the width–depth dynamics of parallel thinking by periodically eliciting intermediate answers from all branches. Our analysis reveals three key insights: non-monotonic scaling across width–depth allocations, heterogeneous reasoning branch lengths, and early stabilization of global consensus. Guided by these insights, we introduce **Parallel-Probe**, a training-free controller designed to optimize online parallel thinking. Parallel-Probe employs consensus-based early stopping to regulate reasoning depth and deviation-based branch pruning to dynamically adjust width. Extensive experiments across three benchmarks and multiple models demonstrate that Parallel-Probe establishes a superior Pareto frontier for test-time scaling. Compared to standard majority voting, it reduces sequential tokens by up to **35.8%** and total token cost by over **25.8%** while maintaining competitive accuracy.

## 1. Introduction

Parallel thinking has emerged as a promising paradigm for improving LLM reasoning by exploring multiple reasoning trajectories in parallel and aggregating them (e.g., via voting, selection, or summarization) (Comanici et al., 2025; Zheng et al., 2025; Wen et al., 2025). By maintaining multiple candidate reasoning trajectories, it reduces the brittleness of single-chain reasoning, where early mistakes can easily compromise the entire reasoning process (Wang et al., 2022a; Zheng et al., 2025). Moreover, parallel thinking is also hardware-friendly: it naturally aligns with modern GPU parallelism, enabling high-throughput batched decoding (Rodionov et al., 2025; Hsu et al., 2025; Yang et al., 2025c). However, this paradigm often requires massive token generation (Fu et al., 2025b), e.g., token usage nearly scales with the number of parallel branches, thereby posing significant challenges to efficiency.

To improve efficiency, previous work studies efficient reasoning at test time. The majority of the research investigates early-stopping strategies for sequential generation (e.g., extended Chain-of-Thought), leveraging signals such as confidence (Fu et al., 2025b), hidden states (Li et al., 2026), or answer convergence (Liu & Wang, 2025; Zhang et al., 2025b). Since these approaches focus on the internal state of individual trajectories, they ignore critical global information across branches (e.g., consensus), making them sub-optimal in parallel thinking settings. Meanwhile, several studies have explored adaptive sampling to reduce the inference cost of self-consistency (Mao et al., 2025; Aggarwal et al., 2023; Wan et al., 2025; Fu et al., 2025b; Huang et al., 2025; Zheng et al., 2026). Since these methods rely on sequential control loops, they transform parallel sampling into a semi-sequential process. Consequently, even though sample efficiency is improved, the increased latency cancels out the speed advantage. Efficient parallel thinking in an **online** setting has received limited attention, particularly the simultaneous launch of multiple paths.

The fundamental challenge lies the intrinsic independence of parallel decoding threads, where each branch evolves without regard for the progression of others. This isolation leads to suboptimal resource allocation and decoding of redundant trajectories. This raises a pivotal question: *Can we introduce lightweight global signals to facilitate efficient, hardware-friendly parallel thinking?*

To bridge this gap, we introduce 2D Probing, a black-box interface that periodically injects an end-of-think token to elicit intermediate answers from each branch during decoding. This constructs a 2D probing matrix with intermediate answers, defined by branch index (width) and probing period

---

[1]Department of Computer Science, University of Maryland, College, Park [2]Washington University in St. Louis [3]University of North Carolina at Chapel Hill. Correspondence to: Tong Zheng <tzheng24@umd.edu>, Heng Huang <heng@umd.edu>.

*Proceedings of the 43rd International Conference on Machine Learning*, Seoul, South Korea. PMLR 306, 2026. Copyright 2026 by the author(s).

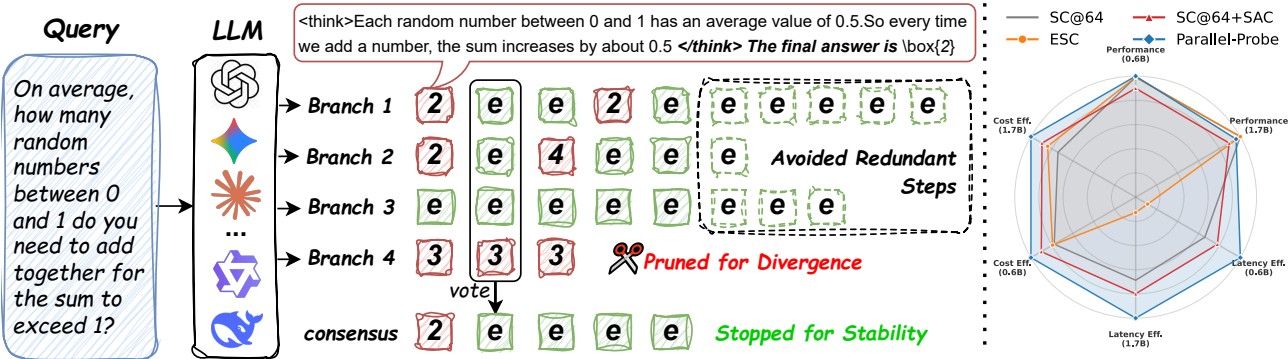

*Figure 1.* **Overview of the Parallel-Probe framework.** It monitors $N$ parallel reasoning branches via continuous 2D probing. **(1) Divergence Pruning:** Outlying trajectories that drift from the global majority (e.g., Branch 4) are aggressively pruned to save compute. **(2) Stability Stopping:** The global controller halts the entire ensemble once the consensus stabilizes, preventing the execution of redundant post-convergence steps (dashed area). Crucially, Parallel-Probe is model-agnostic and compatible with various off-the-shelf LLMs. We evaluate Performance, Cost Efficiency, and Latency Efficiency across 0.6B and 1.7B models. Values are averaged across all datasets and normalized such that the best-performing method on each axis equals 1.0. Parallel-Probe (blue) achieves the largest coverage area, demonstrating a superior balance between high accuracy and computational efficiency compared to SC and ESC methods.

(depth). Such a probing matrix enables fine-grained monitoring of reasoning trajectories. To analyze these dynamics, we develop SCOUT (**S**equential & **C**oncurrent **O**ffline **U**tilization **T**estbed), an evaluation platform designed to rapidly assess different strategies using pre-sampled data.

Using SCOUT, we discover three simple but important insights that explain why standard per-trajectory early stopping is suboptimal for online parallel thinking: (i) Scaling is non-monotonic: Accuracy depends heavily on how width and depth are balanced, not just the total token budget (Figure 2 (a)); (ii) Lengths of reasoning branches are highly uneven (Figure 2 (b) and Figure 7). (iii) Consensus stabilizes early: Early majority votes are often unstable and inaccurate, but they converge to a reliable consensus long before all branches terminate (Figure 2 (c)).

Guided by these insights, we propose Parallel-Probe, a training-free controller designed to optimize online parallel thinking through two complementary mechanisms along both dimensions. This aligns with Insight (i). Figure 1 (left) illustrates the working mechanism. Motivated by Insight (ii) and (iii), we first design **Consensus-based Early Stopping**, which uses the consensus of parallel branches to verify sequential stability, terminating generation once the period-wise majority answer becomes stable. Meanwhile, to further prevent long-tail token waste, we implement **Deviation-based Branch Pruning**, which conversely uses global trends to identify deviating paths, dynamically removing outliers.

We validate Parallel-Probe across three benchmarks and multiple models. The results demonstrate that our method consistently achieves a superior Pareto frontier with better accuracy–efficiency trade-off compared to strong baselines. Specifically, Parallel-Probe reduces sequential tokens, which is a proxy for latency by more than 30% and total token cost by over 20% compared to Self-Consistency (SC) (Wang et al., 2022a), while maintaining competitive accuracy. As illustrated in Figure 1 (right), our approach consistently dominates competing methods across performance, latency-aware efficiency, and cost efficiency dimensions, highlighting the effectiveness of global probing-based control for efficient online parallel thinking.

## 2. 2D Probing: Dynamics and Principles

Standard parallel thinking is not able to observe and utilize its cross-branch trajectory. We address this by introducing 2D probing, which maps parallel thinking traces into a structured matrix $\mathbf{A}$ (Sec. 2.1). Analysis of this matrix reveals a dispersion-to-consensus transition: global majority vote often stabilizes long before the termination of redundant, long-tailed branches (Sec. 2.2). This empirical gap suggests that optimal control requires joint regulation of width and depth based on global consensus rather than local information within each trajectory (Sec. 2.3).

### 2.1. 2D Probing as a Diagnostic Interface

Reasoning paths are generated independently in parallel thinking, lacking cross-branch visibility during generation. Without access to global signals such as consensus or divergence, the system often sustains redundant or outlying trajectories, leading to inefficient resource allocation. To address this problem, we introduce *2D probing*, a lightweight diagnostic interface for parallel decoding that periodically queries intermediate *answer-so-far* states from all parallel thinking branches during inference.

Formally, we periodically intercept each of the $N$ parallel

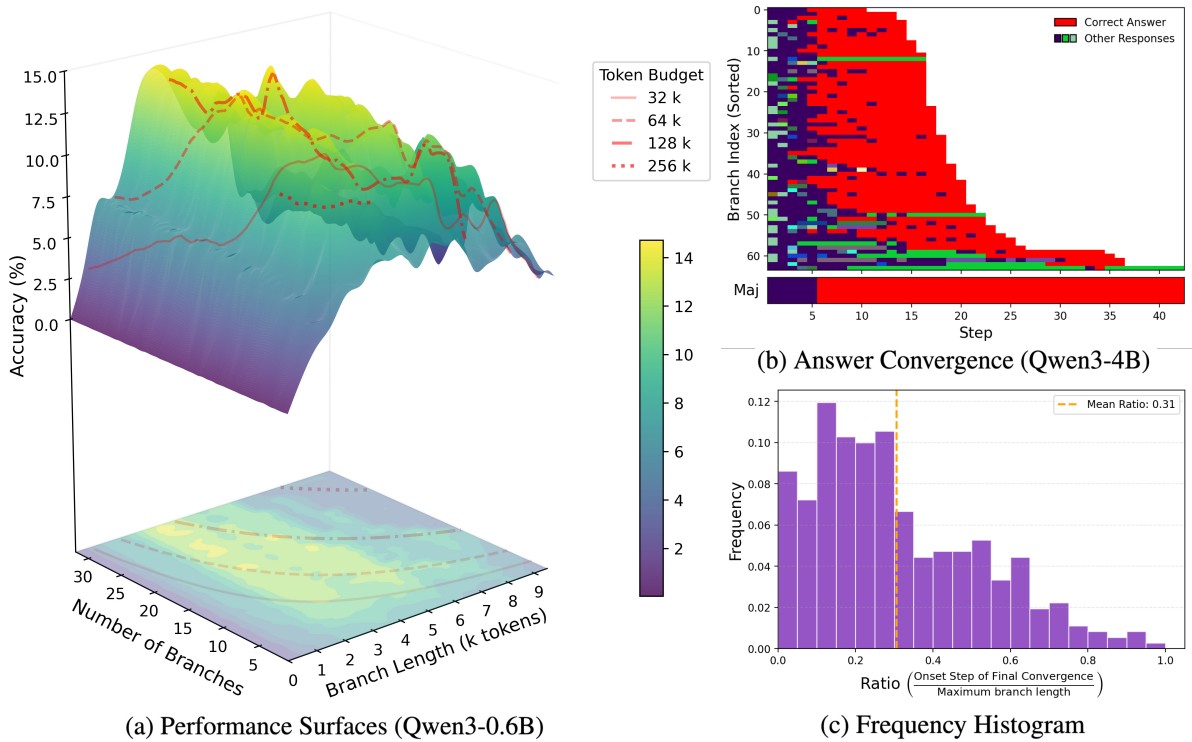

(a) Performance Surfaces (Qwen3-0.6B)

(b) Answer Convergence (Qwen3-4B)

(c) Frequency Histogram

*Figure 2.* **Analysis of Model Performance and Dynamics.** Detailed experimental setups and additional examples for subfigures (a), (b), and (c) are provided in Appendix A. (a) AIME24 performance of Qwen3-0.6B across varying branch numbers and lengths. The accuracy is measured via Majority Voting. Red lines indicate fixed total token budgets (branch length × number of branches), ranging from 32K to 256K. (b) Answer convergence behavior for a representative AIME25 question using Qwen3-4B across different probing steps. Red denotes the group corresponding to the correct answer at each step, while other colors represent distinct incorrect answer groups. (c) Convergence patterns across different models and datasets. We report the convergence onset ratio, defined as the probing step at which the final majority answer first becomes consensus over the maximum branch length.

branches at a fixed probe interval of $\Delta$ tokens. At each probing step $t \in \{1, 2, \ldots, T\}$, we apply an answer-forcing intervention: we append a termination-triggering sequence (e.g.,`</think>` *The final answer is* ) to the current reasoning prefix of each branch. This prompts the model to generate an answer based on the information contained in the existing context. We formalize the probing results as a matrix $\mathbf{A} \in \mathcal{V}^{N \times T}$, where $\mathcal{V}$ denotes all possible answers and $\mathbf{A}_{i,t}$ corresponds to the response of the $i$-th branch at the $t$-th probing step.

### 2.2. Observations From 2D Probing

By analyzing the 2D probing matrix $\mathbf{A}$, we uncover several structural properties of parallel thinking:

**Observation 1: The Non-Monotonicity of Width-Depth Scaling.** By leveraging dense probing traces, we sweep the width–depth scaling space and characterize the performance of a model on a specific dataset as a 3D surface (Figure 2 (a) ). We provide detailed settings and more examples in Appendix A. Our results show that accuracy is not a monotonic

function of either width or depth. Notably, the performance varies substantially across various combinations of chain length and count, even when constrained to the same budget (iso-budget lines). This observation indicates that compute efficiency in parallel thinking is highly sensitive to how budget is distributed across dimensions, rather than the total budget alone.

**Observation 2: The Heterogeneity of Reasoning Branch Lengths.** Analyzing the depth dimension of the 2D probing matrices, we observe that reasoning lengths across parallel branches are highly heterogeneous, exhibiting a long-tailed distribution (Figure 2 (b) and 7). While many branches stabilize or terminate after relatively few decoding steps, a small fraction of branches produce substantially longer reasoning traces. This skewness implies that the total computational cost is often dominated by a few outlying trajectories.

**Observation 3: The Early Stabilization of Global Consensus.** We find that the majority-voting outcome typically reaches a stable equilibrium long before the completion of

the longest reasoning branches. As visualized in the bottom panel of Figure 2(b), the collective decision often stabilizes while several branches are still in the mid-stages of decoding. To quantify this, we measure the convergence onset ratio, which is defined by the step where the final majority answer first emerges relative to the maximum branch length. The distribution in Figure 2(c) shows an average ratio of only 0.31, highlighting the substantial redundancy and token inefficiency inherent in standard parallel decoding.

**The Need for Global Control.** These observations expose a fundamental mismatch in current parallel thinking designs: while reasoning branches are executed independently, the "signal" is a collective, global property. Traditional stopping criteria, which rely on local trajectory signals (e.g., confidence (Fu et al., 2025b) or answer convergency (Zhang et al., 2025b; Liu & Wang, 2025)), fail to capture this cross-branch consensus. Consequently, a new set of principles are required to shift control from individual trajectories to the global dynamics of the parallel thinking.

### 2.3. Principles for Efficient Parallel Control

The empirical findings from our 2D probing analysis directly motivate three core principles for designing efficient parallel thinking systems.

**Principle 1: Joint Optimization of Width and Depth.** Efficiency cannot be achieved by scaling along a single fixed dimension. Effective control must jointly regulate both the number of parallel branches (width) and their generation length (depth), dynamically allocating the token budget to widen the search space or deepen reasoning chains based on real-time difficulty.

**Principle 2: Adaptive Pruning of Divergent Branches.** Identifying and removing outliers is crucial for resource efficiency. Effective control should aggressively prune divergent branches that drift from the emerging global consensus, thereby mitigating the computational waste of long-tail trajectories while preserving the quality of the majority vote.

**Principle 3: Consensus-Driven Early Termination.** The termination condition should be decoupled from individual branch status. Stopping decisions must be governed by the stability of the global consensus, halting the entire parallel ensemble immediately once the majority vote becomes robust, rather than waiting for the slowest branch to finish.

## 3. Parallel-Probe: Online Control for Parallel Thinking via Probing

Based on the above observations and principles, we introduce our approach, Parallel-Probe (Figure 1). It is a training-free online control policy for parallel thinking. Parallel-Probe exploits global convergence signals exposed by 2D probing, and performs budget control jointly along width and depth. Specifically, it manages effective width via deviation-aware branch pruning and regulates effective depth via global, consensus-driven early stopping.

**Consensus-based early stopping.** Guided by the observation that global consensus stabilizes prematurely (Observation 3), Parallel-Probe monitors the probing matrix $\mathbf{A}$ column-wise to detect the onset of convergence.

Let $d_t$ denote the majority consensus at the $t$-th probing step

$$d_t = mode(\mathbf{A}_t),\qquad(1)$$

where $\mathbf{A_t} = [\mathbf{A}_{1,t}, \mathbf{A}_{2,t}, \ldots, \mathbf{A}_{N,t}]^\top$ represents the snapshot of answers across all $N$ branches at time $t$, $mode(\cdot)$ represents majority voting operations. The early stopping policy halts execution at step $T_{\text{stop}}$ if the consensus remains invariant for $u$ consecutive steps:

$$T_{\text{stop}} = \min\{t \geq u | d_t = d_{t-1} = \cdots = d_{t-(u-1)}\}.\quad(2)$$

Utilizing this signal, Parallel-Probe effectively reclaims the compute budget typically wasted on the "long-tail" of reasoning trajectories, as it no longer requires branches to reach their termination once a stable consensus $d_t$ has emerged.

**Deviation-based branch pruning.** While early stopping regulates reason depth, deviation-aware pruning complements this by thinning the reason width. Guided by Principle 2, this mechanism identifies and deactivates branches that significantly diverge from the consensus.

Formally, a branch $i$ is pruned at step $t$ if its output consistently deviates from the consensus within a lookback window of size $k$:

$$\text{Prune branch } i \text{ if } \sum_{j=0}^{k-1} \mathbb{1}(\mathbf{A}_{i,t-j} \neq d_{t-j}) \geq k,\quad(3)$$

where $\mathbb{1}(\cdot)$ is the indicator function.

**Warmup Stage.** To preserve reasoning diversity and prevent the premature deactivation of promising trajectories during their initial development, we implement a warmup stage with $W$ steps. During this phase (where $t < W$), both early stopping and deviation-aware pruning are suppressed.

**Final Prediction.** Parallel-Probe outputs the stable winner when early stopping triggers; otherwise, it returns majority vote among the final answers of the remaining branches upon reaching the maximum budget.

# 4. SCOUT: Sequential & Concurrent Offline Utilization Testbed

To conduct a systematic and efficient investigation of the trade-offs in test-time scaling, we introduce SCOUT. A core design principle of this framework is the disentanglement of reasoning generation from strategy evaluation. Conducting online inference for every possible configuration would be computationally prohibitive and difficult to reproduce. By separating the construction of the reasoning space from the exploration of scaling policies, SCOUT allows us to simulate various strategies with near zero computational overhead.

## 4.1. Data Collection

In the first phase, we construct the static search space, referred to as the candidate pool. For each problem in our benchmark datasets, we sample 128 independent reasoning paths. To capture the dimension of sequential scaling, we employ a probing technique during generation. Specifically, we intervene at fixed intervals of 500 tokens by inserting a specialized termination token (e.g., </think>) to force the model to output a answer based on its current state. This process yields a dense dataset where each trajectory is associated with a series of intermediate answers and their corresponding computational costs. This phase absorbs the entire computational burden of model inference, effectively freezing the available reasoning resources into a static format for downstream analysis.

## 4.2. Simulation Protocol

In the second phase, we utilize the collected data to estimate the performance of various scaling policies. Because the search space is now disentangled from the generation process, we can flexibly simulate diverse strategies—ranging from fixed parallel-sequential configurations to complex, dynamic verification algorithms. For a given policy, we simulate its execution by interacting with the candidate pool. This involves querying paths, checking intermediate answers, and terminating the process based on specific rules. To ensure statistical stability, we repeat this simulation process 64 times for each experimental setting and report the average performance.

Crucially, this disentanglement ensures a strictly fair comparison between our proposed method and other baseline approaches. By evaluating all strategies on subsets drawn from the exact same pool of generated paths, we eliminate the randomness inherent in online generation. This guarantees that any observed performance differences are solely attributable to the logic of the scaling strategy itself, rather than stochastic variations in the model's output.

**Open Source Contribution.** To facilitate future research and ensure reproducibility, we will publicly release both the SCOUT simulation code and part of the dataset of collected reasoning paths.

# 5. Experimental Setups

## 5.1. Models

To evaluate the scalability and generalizability of our proposed framework across models with varying capabilities, we utilize the Qwen-3 model family (Yang et al., 2025a). Specifically, we conduct experiments on four distinct sizes: 0.6B, 1.7B, 4B and 8B. This selection covers a broad spectrum of parameter scales, allowing us to investigate whether the benefits of our joint sequential-parallel scaling strategy persist from lightweight models to more capable ones. All models are evaluated in thinking model.

## 5.2. Evaluation Benchmark

**Datasets.** Since base models already perform very well on standard benchmarks, there is limited room to observe the benefits of test-time scaling. Therefore, we focus on three difficult benchmarks: AIME 2024, AIME 2025, and HMMT 2025 (Balunović et al., 2025). These tasks require complex logic and provide a sufficiently high difficulty level to properly evaluate advanced reasoning capabilities.

**Metrics.** We report performance using three key metrics: (a) **Accuracy**, defined as the percentage of correctly solved problems; (b) **Total Tokens**, the sum of all tokens generated during inference, representing the total computational cost; and (c) **Sequential Tokens**, which measures the length of the critical path (i.e., the number of tokens in the longest sequential chain). The latter is crucial for capturing real-world latency, as it distinguishes methods that effectively parallelize operations from those that unnecessarily serialize them—specifically, fewer sequential tokens imply better parallel efficiency even when total token consumption is identical.

## 5.3. Baseline Methods

To evaluate the effectiveness of Parallel-Probe, we compare it with representative test-time scaling baselines spanning sequential, parallel, and hybrid settings:

- **SC@64** (Self-Consistency (Wang et al., 2022a)): A standard parallel baseline that samples $N = 64$ independent reasoning trajectories and returns the majority-voted answer.

- **ASC** (Adaptive Self-Consistency (Aggarwal et al., 2023)): An adaptive parallel method that incrementally samples trajectories and stops once a predefined

*Table 1.* **Comparison of efficient reasoning approaches across three benchmarks.** Acc. denotes accuracy. SeqToks measures the latency-critical sequential tokens on the critical path (i.e., the maximum number of generated tokens among all branches for parallel methods, and the total generated tokens for sequential methods), while Tokens counts the total generated tokens summed over all branches (i.e., overall inference cost). Lower is better for both SeqToks and Tokens.

| Method | Type | AIME24 | | | AIME25 | | | HMMT25 | | | Avg. | | |
|---|---|---|---|---|---|---|---|---|---|---|---|---|---|
| | | Acc. ↑ | SeqTokens ↓ | Tokens ↓ | Acc. ↑ | SeqTokens ↓ | Tokens ↓ | Acc. ↑ | SeqTokens ↓ | Tokens ↓ | Acc. ↑ | SeqTokens ↓ | Tokens ↓ |
| *Base Model: Qwen3-0.6B* | | | | | | | | | | | | | |
| SC@64 | Parallel | 21.4 | 32.7k | 1008.6k | 28.9 | 31.1k | 890.5k | 18.1 | 31.0k | 937.8k | 22.8 | 31.6k | 945.7k |
| ASC | Seq. | 21.4 | 805.5k | 805.5k | 28.9 | 653.8k | 653.8k | 18.1 | 580.8k | 580.8k | 22.8 | 680.0k (+2051.1%) | 680.0k (-28.1%) |
| ESC | Hybrid | 21.4 | 192.9k | 986.7k | 28.9 | 171.5k | 868.8k | 18.1 | 179.5k | 923.9k | 22.8 | 181.4k (+473.9%) | 926.5k (-2.0%) |
| SC@64 + SAC | Parallel | 19.5 | 26.8k | 820.7k | 25.4 | 27.2k | 819.4k | 17.4 | 26.3k | 808.2k | 20.7 | 26.8k (-15.3%) | 816.1k (-13.7%) |
| Parallel-Probe | Parallel | 21.8 | 20.8k | 773.8k | 29.7 | 19.6k | 697.8k | 18.5 | 20.5k | 734.5k | 23.3 | 20.3k (-35.8%) | 735.3k (-22.2%) |
| *Base Model: Qwen3-1.7B* | | | | | | | | | | | | | |
| SC@64 | Parallel | 72.5 | 31.4k | 1025.8k | 44.4 | 30.0k | 1054.1k | 24.2 | 32.4k | 1132.9k | 47.0 | 31.3k | 1070.9k |
| ASC | Seq. | 72.3 | 482.6k | 482.6k | 44.4 | 600.9k | 600.9k | 24.2 | 586.3k | 586.3k | 47.0 | 556.6k (+1679.7%) | 556.6k (-48.0%) |
| ESC | Hybrid | 72.5 | 170.4k | 909.2k | 44.4 | 160.6k | 913.8k | 24.2 | 174.9k | 1014.2k | 47.0 | 168.6k (+439.2%) | 945.7k (-11.7%) |
| SC@64 + SAC | Parallel | 64.5 | 27.3k | 868.2k | 40.0 | 26.4k | 909.0k | 21.4 | 26.9k | 889.1k | 42.0 | 26.9k (-14.1%) | 888.8k (-17.0%) |
| Parallel-Probe | Parallel | 68.1 | 20.5k | 748.5k | 44.7 | 21.3k | 775.8k | 22.6 | 22.8k | 860.2k | 45.1 | 21.5k (-31.3%) | 794.8k (-25.8%) |
| *Base Model: Qwen3-4B* | | | | | | | | | | | | | |
| SC@64 | Parallel | 80.0 | 29.3k | 886.8k | 76.6 | 30.5k | 1088.1k | 43.6 | 33.9k | 1168.3k | 66.8 | 31.2k | 1047.7k |
| ASC | Seq. | 80.0 | 214.2k | 214.2k | 76.6 | 325.1k | 325.1k | 43.6 | 487.3k | 487.3k | 66.7 | 342.2k (+995.4%) | 342.2k (-67.3%) |
| ESC | Hybrid | 80.0 | 98.9k | 528.9k | 76.6 | 137.0k | 793.3k | 43.6 | 174.0k | 990.2k | 66.8 | 136.6k (+337.3%) | 770.8k (-26.4%) |
| SC@64 + SAC | Parallel | 80.0 | 24.8k | 782.2k | 73.3 | 27.9k | 995.4k | 41.9 | 27.1k | 863.0k | 65.1 | 26.6k (-14.8%) | 880.2k (-16.0%) |
| Parallel-Probe | Parallel | 79.7 | 19.2k | 688.9k | 76.1 | 22.2k | 806.0k | 44.7 | 21.5k | 872.3k | 66.8 | 20.9k (-33.0%) | 789.0k (-24.7%) |
| *Base Model: Qwen3-8B* | | | | | | | | | | | | | |
| SC@64 | Parallel | 80.4 | 30.1k | 910.8k | 76.7 | 30.7k | 1124.4k | 48.9 | 34.8k | 1267.0k | 68.6 | 31.9k | 1100.7k |
| ASC | Seq. | 80.4 | 226.0k | 226.0k | 76.7 | 406.2k | 406.2k | 48.8 | 565.1k | 565.1k | 68.6 | 399.1k (+1152.7%) | 399.1k (-63.7%) |
| ESC | Hybrid | 80.4 | 84.7k | 459.4k | 76.7 | 132.4k | 793.1k | 48.6 | 184.5k | 1062.1k | 68.6 | 133.9k (+320.1%) | 771.5k (-29.9%) |
| SC@64 + SAC | Parallel | 76.7 | 25.6k | 773.4k | 70.2 | 28.1k | 998.5k | 42.7 | 28.5k | 896.8k | 63.2 | 27.4k (-14.0%) | 889.5k (-19.2%) |
| Parallel-Probe | Parallel | 81.5 | 20.3k | 730.8k | 76.9 | 21.9k | 846.7k | 47.1 | 22.4k | 897.2k | 68.5 | 21.6k (-32.3%) | 824.9k (-25.1%) |

consensus threshold is reached. We follow the original setting with threshold 0.95.

- **ESC** (Early Stopping Consistency (Li et al., 2024)): A chunk-based hybrid approach that generates trajectories in parallel and terminates early when answer stability is detected within a sliding window. We use a chunk size of 8.

- **SC@64 + SAC** (Liu & Wang, 2025): A baseline that applies SAC as a trajectory-level early stopping rule within SC, terminating each trajectory upon local answer convergence before majority voting.

## 6. Results and Analysis

### 6.1. Main Results

Table 1 summarizes the overall performance of Parallel-Probe against representative efficient reasoning baselines across three benchmarks and four foundation models.

Overall, Parallel-Probe consistently achieves a better accuracy–efficiency trade-off than strong baselines.

- Compared to the standard SC@64 baseline, Parallel-Probe substantially reduces computation (both sequential tokens and total tokens, e.g., more than 30% and 20% respectively) while largely preserving accuracy.

- Despite existing efficient parallel sampling techniques,

e.g., ASC and ESC can effectively cut down total token usage, they always suffer an increased usage of sequential tokens. This is due to their sequential control. **By contrast, our Parallel-Probe does not rely on such sequential control and can effectively reduce the usage of both sequential tokens and total token.**

- When applying existing early-stopping approaches to the parallel thinking setting, they reduce sequential and total token usage by over 10%, but at the cost of a substantial performance drop, e.g., from 68.6 to 63.2 on Qwen3-8B. In contrast, our Parallel-Probe maintains competitive performance compared to the SC@64 baseline while achieving larger computational reductions in both sequential and total token consumption. This difference highlights that directly extending early-stopping approaches originally designed for sequential thinking to parallel thinking is sub-optimal, due to the lack of global control signals.

### 6.2. Scaling with Inference Budget

Figure 3 illustrates the test-time scaling behavior under different inference budgets, where the x-axis denotes token cost (log-scale) and the y-axis denotes accuracy. We compare our Parallel-Probe with both SC and SC + ASC across two Qwen3 model sizes (0.6B and 1.7B) on AIME24 and AIME25. Overall, Parallel-Probe achieves a superior Pareto frontier for test-time scaling. Notably, SC + ASC, which

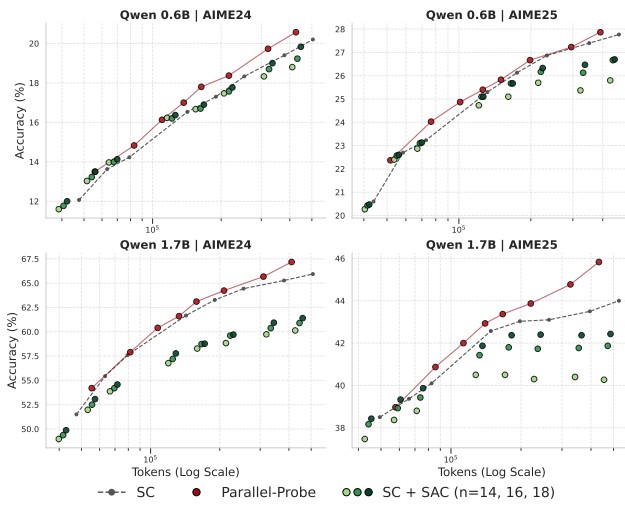

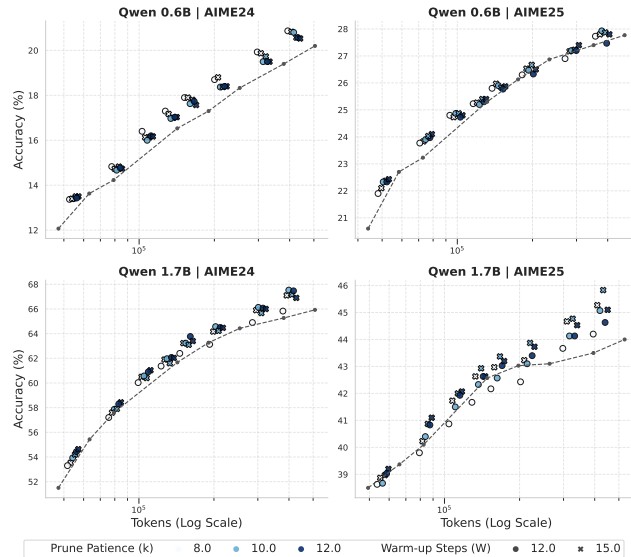

*Figure 3.* Accuracy–token scaling curves comparing the SC, SC+SAC, and our Parallel-Probe across different models and benchmarks. Notably, we show the results of SC+SAC under three different settings ($n=14$, $n=16$, $n=18$). The x-axis is shown in log scale. Parallel-Probe consistently achieves higher accuracy under the same or lower token budget.

*Figure 4.* Hyper-parameter sensitivity analysis of Parallel-Probe under different prune patience $k$ and warm-up steps $W$ on Qwen-0.6B and Qwen-1.7B across AIME24 and AIME25.

only considers per-trajectory information, fails to achieve effective and efficient parallel thinking. Under three different hyper-parameter setups, SC + ASC consistently achieves poorer performance compared SC. This validated our observations in Sec 2.2 that there is still some token inefficiency when only considering per-trajectory information.

### 6.3. Ablation Studies

We conduct ablation studies on Qwen-3-0.6B to examine the contribution of each component in Parallel-Probe. Table 2 reports results on AIME24 and AIME25 in terms of accuracy, sequential tokens, and total token usage.

When removing the global 2D probing signals, which degrades our method to a local early-stopping strategy (SC + SAC), the overall performance drops substantially, with average accuracy decreasing from 25.8 to 22.4. Meanwhile, both sequential and total token costs increase by 33.7% and 11.4%, respectively. This demonstrates that fine-grained global probing information is crucial for deriving reliable control signals and achieving efficient parallel reasoning.

Disabling the proposed deviation-based pruning leads to significantly higher computational cost while achieving comparable accuracy. Specifically, the method requires 4.7% more sequential tokens and 14.7% more total tokens on average. This confirms that pruning unpromising branches based on deviation dynamics is essential for reducing redundant computation in parallel reasoning.

When the consensus-based early stopping mechanism is removed, the performance remains largely unchanged, but

with an increased token usage up to 13.1% and 8.6%, respectively.

Finally, removing the warmup stage degrades performance, with average accuracy dropping from 25.8 to 23.5, despite reducing sequential and total tokens by 2.9% and 19.2%, respectively. This suggests that applying probing-guided control too early based on unstable signals leads to suboptimal pruning and early stopping decisions.

### 6.4. Hyperparameter Sensitivity

We further conduct hyperparameter sensitivity analysis on Parallel-Probe. Specifically, we study the pruning tolerance $k$ and the warm-up length $W$. We evaluate $k \in \{8, 10, 12\}$ and $W \in \{12, 15\}$ on Qwen-3-0.6B and Qwen-3-1.7B across AIME24 and AIME25. As shown in Figure 4, varying these hyperparameters mainly moves the operating point of Parallel-Probe along a consistent accuracy–token trade-off curve, which remains systematically above the SC baseline curve (as shown in the dotted lines in Figure 4). This indicates that Parallel-Probe robustly achieves superior efficiency–accuracy trade-offs and is not sensitive to hyperparameter choices within the examined ranges.

## 7. Related Work

### 7.1. Efficient Parallel Reasoning

To mitigate the computational cost of fixed-budget search, recent research focuses on dynamic resource allocation. Aggarwal et al. (2023) and Li et al. (2024) propose adaptive mechanisms that halt generation once a consensus threshold

*Table 2.* Ablation study of Parallel-Probe on two benchmarks. We report Accuracy, sequential tokens (SeqTok; lower is better), and total generated tokens (TotTok; lower is better). Δ reports the relative change compared to Parallel-Probe (negative means fewer tokens / lower cost).

| Ablation | AIME24 | | | AIME25 | | | Avg. | | |
|---|---|---|---|---|---|---|---|---|---|
| | Acc. | SeqTok | TotTok | Acc. | SeqTok | TotTok | Acc. | SeqTok | TotTok |
| **Parallel-Probe (Full)** | 21.8 | 20.8k | 773.8k | 29.7 | 19.6k | 697.8k | 25.8 | 20.2k | 735.8k |
| w/o deviation-based branch pruning | 21.1 | 21.3k$^{(+2.3\%)}$ | 872.0k$^{(+12.7\%)}$ | 26.8 | 21.0k$^{(+7.3\%)}$ | 815.4k$^{(+16.9\%)}$ | 23.9 | 21.1k$^{(+4.7\%)}$ | 843.7k$^{(+14.7\%)}$ |
| w/o consensus-based early stopping | 22.0 | 23.8k$^{(+14.2\%)}$ | 842.8k$^{(+8.9\%)}$ | 29.0 | 21.9k$^{(+11.8\%)}$ | 754.6k$^{(+8.1\%)}$ | 25.5 | 22.8k$^{(+13.1\%)}$ | 798.7k$^{(+8.6\%)}$ |
| w/o warmup stage | 18.3 | 20.6k$^{(-1.3\%)}$ | 633.5k$^{(-18.1\%)}$ | 28.7 | 18.7k$^{(-4.5\%)}$ | 555.9k$^{(-20.3\%)}$ | 23.5 | 19.6k$^{(-2.9\%)}$ | 594.7k$^{(-19.2\%)}$ |
| w/o leveraging 2d probing information | 19.5 | 26.8k$^{(+28.8\%)}$ | 820.7k$^{(+6.1\%)}$ | 25.4 | 27.2k$^{(+38.9\%)}$ | 819.4k$^{(+17.4\%)}$ | 22.4 | 27.0k$^{(+33.7\%)}$ | 820.0k$^{(+11.4\%)}$ |

is met, while Wang et al. (2025a) further optimizes efficiency by allocating samples based on query difficulty. Beyond count reduction, confidence-aware approaches weight reasoning paths to identify high-quality solutions with fewer samples (Huang et al., 2025; Taubenfeld et al., 2025; Fu et al., 2025b). However, they predominantly adopt sequential sampling to obtain these samples, limiting the hardware efficiency of parallel thinking. More recently, fine-grained methods like Dynamic Self-Consistency (Wan et al., 2025), Self-Truncation (Wang et al., 2025c), DeepPrune (Tu et al., 2025), Step (Liang et al., 2026) and Slim-SC (Hong et al., 2025) prune unpromising trajectories mid-generation to minimize wasteful computation on incorrect paths. Despite their effectiveness, these methods lack principled modeling of the global dynamics across parallel reasoning trajectories, resulting in coarse-grained control over parallel thinking.

### 7.2. Efficient Sequential Reasoning

To optimize the depth of thought without additional training, recent research focuses on dynamic early exiting mechanisms. A primary strategy involves monitoring uncertainty metrics: Wang et al. (2025b) and Sharma & Chopra (2025) utilize entropy after the reasoning block or at the sequence level as confidence signals, while Yong et al. (2025) estimates this empirically via multiple rollouts or beam search. Alternatively, termination decisions can be guided by output stability, using answer convergence across steps to signal sufficiency (Liu & Wang, 2025; Mao et al., 2025; Fu et al., 2025a; Zhang et al., 2025b). Beyond output statistics, Zhang et al. (2025a) suggest probing hidden states directly for self-verification, allowing models to halt inference once an internal correctness threshold is met (Yang et al., 2025b). Despite their success in efficient sequential reasoning, these methods fail to leverage the global dynamics of parallel thinking (as reflected in our Observations 1–3). As a result, directly applying them to parallel reasoning settings is sub-optimal.

### 7.3. Test-Time Scaling

To optimize the efficiency of complex reasoning, recent studies have shifted focus toward the strategic allocation of test-time computation (Snell et al., 2024; Chen et al., 2025b). A primary manifestation of this trend is the use of tree-search frameworks, which aggregate diverse reasoning paths and employ sparse activation to manage complexity (Bi et al., 2024; Lample et al., 2022; Koh et al., 2024; Zheng et al., 2025). To further refine these search spaces, step-wise verifiers have become essential for dynamically pruning unproductive branches (Wang et al., 2022b; Li et al., 2022; Lightman et al., 2023). Beyond search-level optimizations, performance can be bolstered by diversifying query formulations (Huang et al., 2024) or through iterative refinement cycles that bootstrap the model's self-correction capabilities to handle increasingly intricate tasks (Chen et al., 2025a; Welleck et al., 2022; Madaan et al., 2023; Aggarwal et al., 2024). Our work leverages global dynamic signals from black-box 2D probing to enable principled control along both depth and width dimensions.

## 8. Conclusion

We investigated how to make parallel thinking in LLMs more efficient. By introducing 2D probing, a black-box interface that monitors reasoning trajectories across both width and depth, we identified several hidden dynamics: non-monotonic scaling, early consensus, and highly varied branch lengths. These findings suggest that standard early-stopping strategies which only leverage information within each trajectories are insufficient for managing parallel thinking. Guided by these insights, we proposed Parallel-Probe, a training-free online controller that leverages global probing signals to dynamically coordinate parallel generation via deviation-based branch pruning and consensus-based early stopping. To facilitate principled evaluation of parallel thinking strategies, we further introduce SCOUT, an offline testbed that decouples generation from control, enabling rapid exploration of diverse width–depth configurations and efficiency–accuracy trade-offs. Extensive experiments across multiple model scales and challenging reasoning benchmarks demonstrate that Parallel-Probe consistently achieves superior Pareto frontiers compared to strong sequential and parallel baselines.

## Acknowledgements

This work was partially supported by NSF IIS 2347592, 2348169, DBI 2405416, CCF 2348306, CNS 2347617, RISE 2536663.

## Impact Statement

We believe this work establishes 2D probing as a powerful interface for understanding and controlling parallel reasoning, and opens a new research direction toward principled, efficient parallel thinking of large language models. Future work may explore learning-based controllers, richer probing signals, and tighter integration between training-time objectives and online parallel control. We will opensource both the code and data of SCOUT to make it easier and more efficient for researchers to explore this direction.

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

# A. Detailed experimental setups and addtional results.

## A.1. Experimental setups of Figure 2(a)

For each dataset–model pair, we collect 128 responses per question. Because response lengths vary significantly, each question induces a irregular shaped majority-voting matrix, as illustrated in Figure 2(b). In the early stages of generation, majority voting can be computed using a large number of branches; however, the number of available branches decreases as responses lengthen, as fewer sequences reach those higher token counts.

As shown in Figure 5, we map the "coverage"—the total number of questions contributing data to each (length, width) coordinate. We observe that coverage becomes increasingly sparse at greater lengths, reflecting the model's tendency to produce substantially longer responses for some queries than for others. To mitigate the potential bias introduced by this uneven distribution, we restrict our primary analysis to the sub-matrix highlighted by the red box, where coverage remains high and consistent across the dataset. We then average the majority-voting accuracy within this stable region to derive a reliable estimate of performance. Results for additional models and datasets are detailed in Figure 6.

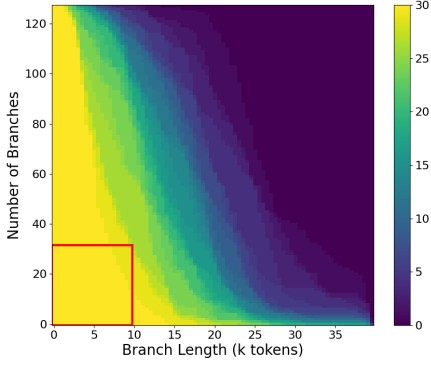

*Figure 5.* Coverage density across varying branch counts and lengths (Qwen3-0.6B, AIME25). Colors indicate the volume of questions with available majority-voting results. The red box highlights the high-coverage region used to mitigate bias from uneven response lengths during accuracy estimation.

## A.2. Experimental setups of Figure 2(b)

Figure 2(b) illustrates the convergence behavior of 64 responses to a representative AIME25 question using Qwen3-4B across various probing steps. While the correct answer is 117, red pixels indicate instances where a probing step yields this correct result. Other colors denote distinct groups of incorrect responses; for example, green represent answer of 101, respectively. We provide additional examples following the same visualization logic in Figure 7.

## A.3. Experimental setups of Figure 2(c)

Figure 2(c) illustrates the distribution of convergence ratios across four model scales (Qwen3-0.6B, 1.7B, 4B, and 8B) evaluated on the AIME24, AIME25, and HMMT25 benchmarks. For each model-dataset pair, we generated 128 independent reasoning trajectories, totaling 360 unique evaluation instances. We define the onset of final convergence as the earliest step at which the majority-vote consensus stabilizes and remains unchanged until the end of the sequence. For each instance, we calculate a ratio by dividing this onset step by the maximum trajectory length within its respective 128-sample set. The histogram presents the frequency distribution of these 360 ratios.

## A.4. Generalization Experiments

Table 4 and 5 demonstrates generalization beyond Qwen model family and mathematical reasoning tasks.

## A.5. Additional Results of Online Evaluation

To directly address the practicality concern, we additionally perform online wall-clock latency evaluation on AIME25 with Qwen-3-0.6B, Qwen-3-4B and Qwen-3-8B.

**Setup.** 4×H200 GPUs, vLLM inference, tensor parallelism = 4 for Qwen family while tensor parallelism = 2 for Phi-4-Reasoning model.

**Metric.** We run questions one by one, record the end-to-end wall-clock time for each question, and report the average latency across the evaluation set.

*Table 3.* Online wall-clock latency evaluation on AIME25

| Method | Time (s/problem) | Acc. |
|---|---|---|
| *Qwen-3-0.6B* | | |
| SC@64 | 170.0 | 28.3 |
| SAC + SC@64 | 163.4 | 25.0 |
| Parallel-Probe | **113.2** | **30.0** |
| *Qwen-3-4B* | | |
| SC@64 | 260.2 | 76.7 |
| SAC + SC@64 | 222.6 | 70.0 |
| Parallel-Probe | **155.8** | **76.7** |
| *Qwen-3-8B* | | |
| SC@64 | 273.1 | 76.7 |
| SAC + SC@64 | 227.9 | 70.0 |
| Parallel-Probe | **196.9** | **76.7** |

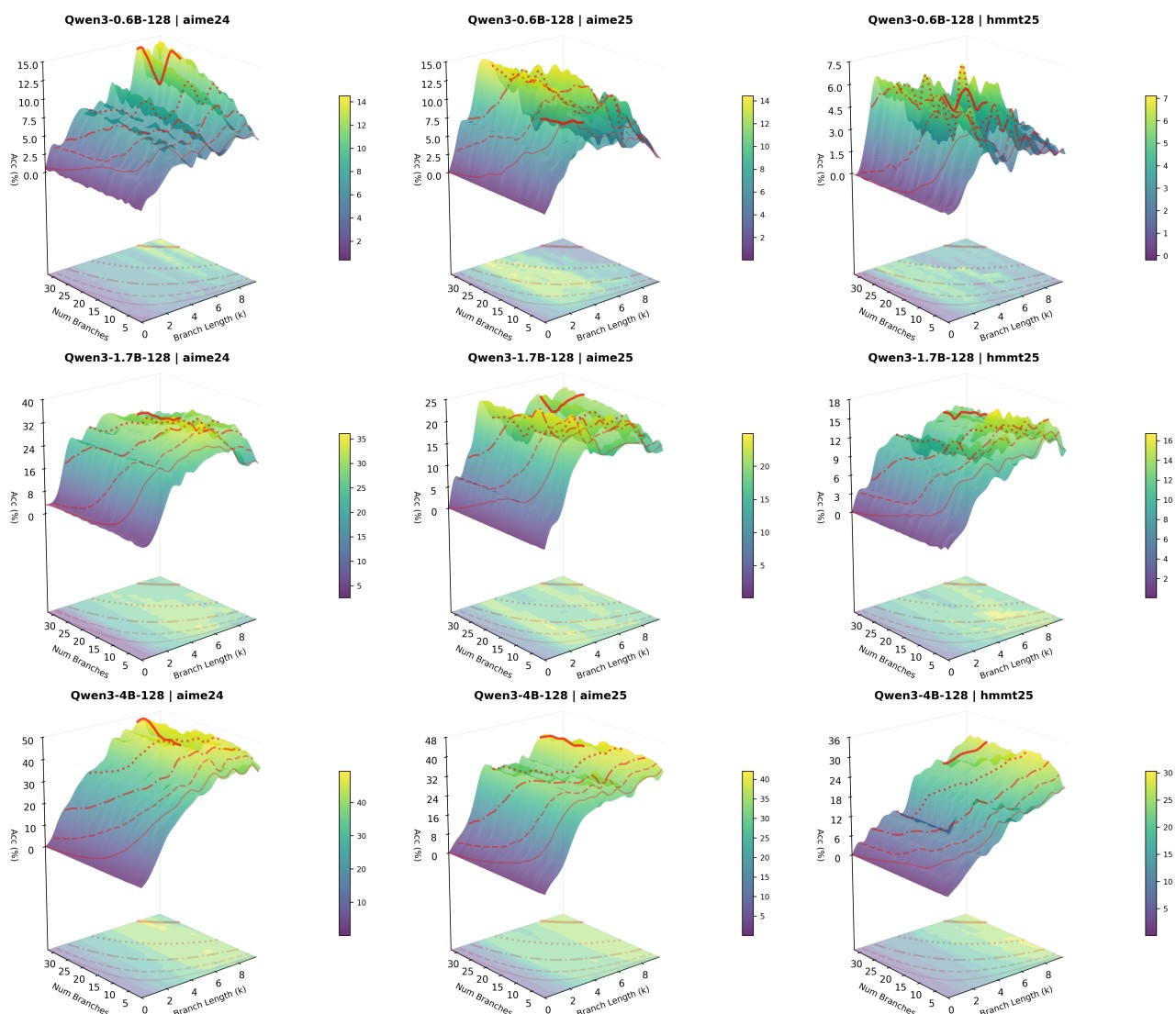

*Figure 6.* Majority voting accuracy with varying branch number and branch lengths across datasets and models.

Table 3 show the results on AIME25. We can find: 1) Parallel-Probe is practically efficient. In our experiments, it achieves lower average wall-clock time than the SC@64 baseline while maintaining comparable performance; and 2) SAC + SC@64 is also faster than SC@64, but remains slower than Parallel-Probe. This suggests that leveraging cross-branch signals enables more effective early stopping and pruning.

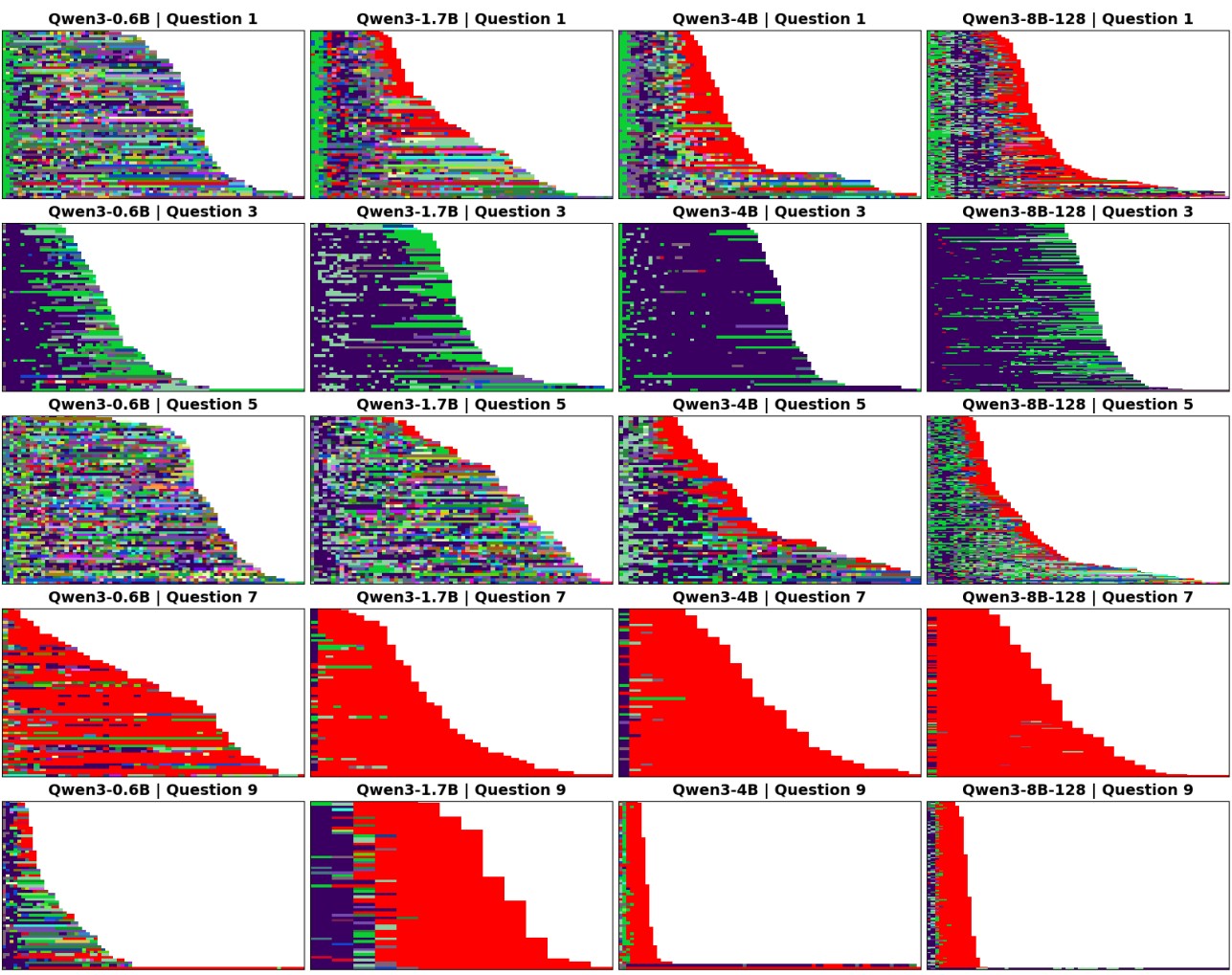

*Figure 7.* Visualization of continuous 2D probing dynamics for parallel reasoning on multiple examples.

*Table 4.* Comparison of SC@64 and Parallel-Probe across models and benchmarks

| Method | Total Tokens | Sequential Tokens | Acc. |
|---|---|---|---|
| *Deepseek-LLaMA-8B on AIME25* | | | |
| SC@64 | 919.5k | 31.4k | 53.3 |
| Parallel-Probe | 742.5k (-19.2%) | 21.8k (-30.6%) | 53.3 |
| *Deepseek-LLaMA-8B on AIME24* | | | |
| SC@64 | 894.7k | 34.5k | 79.2 |
| Parallel-Probe | 662.8k (-25.9%) | 18.7k (-45.8%) | 80.7 |
| *Deepseek-LLaMA-8B on HMMT25* | | | |
| SC@64 | 985.7k | 36.7k | 26.7 |
| Parallel-Probe | 568.3k (-42.3%) | 21.1k (-42.5%) | 27.1 |
| *Phi-4 Reasoning on AIME25* | | | |
| SC@64 | 734.6k | 24.7k | 76.7 |
| Parallel-Probe | 524.7k (-29.8%) | 17.6k (-28.7%) | 76.7 |
| *Qwen-3-4B-Instruct on AIME25* | | | |
| SC@64 | 509.4k | 25.8k | 65.0 |
| Parallel-Probe | 408.4k (-19.8%) | 11.1k (-57.0%) | 65.0 |

*Table 5.* Comparison of methods on Qwen-3-0.6B across additional benchmarks

| Method | Type | Acc.↑ | SeqTokens↓ | Total Tokens↓ |
|---|---|---|---|---|
| *Qwen-3-0.6B on Reasoning-Gym* | | | | |
| SC @ 64 | Parallel | 57.2 | 17.2k | 320.1k |
| ASC | Seq. | 57.4 | 153.5k (+793.6%) | 153.5k (-52.1%) |
| ESC | Hybrid | 57.3 | 62.9k (+266.2%) | 252.4k (-21.1%) |
| SC @ 64 + SAC | Parallel | 57.1 | 11.2k (-34.6%) | 290.7k (-9.2%) |
| Parallel-Probe | Parallel | 57.2 | 6.5k (-62.2%) | 267.7k (-16.4%) |
| *Qwen-3-0.6B on GPQA-Diamond* | | | | |
| SC @ 64 | Parallel | 28.7 | 15.4k | 339.4k |
| ASC | Seq. | 28.7 | 157.3k (+919.7%) | 157.3k (-53.7%) |
| ESC | Hybrid | 28.7 | 63.8k (+313.5%) | 302.2k (-11.0%) |
| SC @ 64 + SAC | Parallel | 26.4 | 10.7k (-30.9%) | 317.3k (-6.5%) |
| Parallel-Probe | Parallel | 30.1 | 7.6k (-50.7%) | 279.6k (-17.6%) |

