# OpenReview forum: "Parallel-Probe: Towards Efficient Parallel Thinking via 2D Probing"
_ICML.cc/2026/Conference — ICML 2026 regular_

### Official Review · Reviewer_oM9s · 2026-03-04

**Soundness:** 3
**Presentation:** 4
**Significance:** 3
**Originality:** 4
**Overall Recommendation:** 5
**Confidence:** 4

**Summary:**

This paper targets efficiency issues in parallel test-time scaling for LLM reasoning, where multiple parallel branches improve accuracy but incur large token and latency costs. The authors propose Parallel-Probe, a training-free controller built on a 2D probing interface that periodically queries intermediate “answer-so-far” states across branches to track global consensus. Using these signals, the method performs consensus-based early stopping and deviation-based branch pruning to reduce redundant computation while maintaining accuracy. Experiments on hard math benchmarks (AIME’24/’25, HMMT’25) across several Qwen3 model sizes report improved accuracy–efficiency trade-offs compared to standard self-consistency and adaptive baselines, and the paper introduces an offline simulation framework (SCOUT) to evaluate strategies more stably.

**Compliance With Llm Reviewing Policy:**

Affirmed.

**Final Justification:**

I think this paper has sufficient novelty, the experiments are solid, and the authors' rebuttal has addressed all my concerns. Therefore, I lean toward clear accept (Overall Recommendation 4 → 5).

**Key Questions For Authors:**

See Weaknesses. I would be willing to raise my score if the authors can address my concerns.

**Limitations:**

yes

**Strengths And Weaknesses:**

## Strengths

* **Addresses a timely and practical problem (significance).** Efficient test-time scaling for reasoning is highly relevant; the paper focuses specifically on *online parallel thinking* efficiency, where naive self-consistency is both expensive and potentially latency-inefficient.

* **A clear diagnostic lens for parallel reasoning (originality / insight).** The proposed 2D probing matrix is a simple but conceptually useful interface to expose width–depth dynamics and global consensus behavior, which prior per-trajectory stopping rules tend to ignore. The three observations (non-monotonic scaling, long-tail lengths, early consensus) are intuitive yet valuable when made concrete and measured.

* **Well-designed evaluation protocol (soundness / practicality).** The proposed SCOUT evaluation framework is a reasonable and careful design choice: by simulating strategies on a pre-collected trajectory pool, it reduces the apparent performance fluctuations that can arise from small benchmark sizes and high sampling variance, making the comparative conclusions more statistically credible.

* **Strong writing and logical flow (presentation / methodology).** The paper is clearly written and well organized. The motivation → probing analysis → algorithm design pipeline is presented in a coherent way that substantially lowers the cognitive load for readers/reviewers.

---

## Weaknesses

* **Missing implementation details and latency-relevant reporting.** Important experimental details are underspecified. For example, it is unclear how **total tokens** and **sequential tokens** are computed for Parallel-Probe, specifically whether the additional probe rollouts are counted in these budgets. In addition, the paper does not clearly describe how probing is executed in a real online system (e.g., whether each probe interrupts ongoing decoding and triggers extra forward passes), which could materially affect end-to-end latency. Reporting a wall-clock latency metric (or an operational-level latency estimate) would strengthen the practicality claims.

* **Limited model diversity (significance / generality).** Experiments are restricted to the Qwen3 family. While these are stress tests for reasoning, it is unclear how the conclusions transfer to other model families (e.g., Llama or Phi).

* **Minor typo / presentation issues.** In Section 6.2 Scaling with Inference Budget, the baseline label “**SC + ASC**” appears to be a typo and should be “**SC + SAC**”. In addition, it would improve clarity if the authors also include the **ASC** and **ESC** baselines in Figure 3 for easier visual comparison.

---

> ### Author Rebuttal · Authors · 2026-03-31
>
> We sincerely thank the reviewer for constructive suggestions and recognizing our contributions. Below, we address the reviewer's specific questions.
>
> **1. Regarding implementation details and latency-related evaluation**
>
> We have clarified the details of token counting and online execution of Parallel-Probe, and further added online wall-clock latency experiments. These results show that Parallel-Probe is not only token-efficient, but also practically efficient in real online serving settings. The details are given below.
>
> **1.1 Token counting for Parallel-Probe**
>
> **We counted probing rollouts in the token budget**. For a sample with **N** parallel branches, let r_i be the number of reasoning tokens generated by branch i, and p_i be the number of probing tokens generated for that branch across all probe steps. Then:
>
> Total Tokens = \sum*{i=1}^{N} (r_i + p_i)
>
> Sequential Tokens = \max*{i=1,\dots,N} (r_i + p_i)
>
> We will make this explicit in the paper.
>
> **1.2 How probing is executed in an online system**
>
> We implemented Parallel-Probe via **looped** **chunked parallel decoding**: all active branches are decoded in parallel for a fixed number of tokens (e.g., 500 tokens), after which we perform a **batched probe** on the active branches. For reasoning models, the probe can be implemented by appending a short answer-extraction suffix (e.g., </think> The final answer is ). Typically the probe is only a very short continuation from the current branch state and this process can also leverage the KV cache, avoiding redundant prefilling.
>
> The extra probing overhead does not necessarily increase end-to-end latency: it can be offset by the ability to terminate disproportionately long or off-consensus branches early, thereby reducing the overall completion time.
>
> **1.3 Online Evaluation of Parallel-Probe and Its Latency**
>
> To directly address the practicality concern, we additionally perform **online wall-clock latency evaluation** on AIME25 with Qwen-3-0.6B, Qwen-3-4B and Qwen-3-8B.
>
> **Setup.** 4×H200 GPUs, vLLM inference, tensor parallelism = 4 for Qwen family while tensor parallelism = 2 for Phi-4-Reasoning model.
> **Metric.** We run questions one by one, record the end-to-end wall-clock time for each question, and report the average latency across the evaluation set.
>
> We will include these latency results in the revision. They show that Parallel-Probe reduces token usage while also improving practical end-to-end latency in the online setting.
>
> | **Method** | **Wall-clock time (s/problem)** | **AIME25 Acc.** |
> | --- | --- | --- |
> | **Qwen-3-0.6B** |  |  |
> | SC@64 | 170.0 | 28.3 |
> | SAC + SC@64 | 163.4 | 25.0 |
> | Parallel-Probe | **113.2** | 30.0 |
> | **Qwen-3-4B** |  |  |
> | SC@64 | 260.2 | 76.7 |
> | SAC + SC@64 | 222.6 | 70.0 |
> | Parallel-Probe | **155.8** | 76.7 |
> | **Qwen-3-8B** |  |  |
> | SC@64 | 273.1 | 76.7 |
> | SAC + SC@64 | 227.9 | 70.0 |
> | Parallel-Probe | **196.9** | 76.7 |
>
> **2. Regarding cross-model family evaluation**
>
> To directly address the concern about model-family generalization, we have added new experiments on **two additional non-Qwen reasoning models**, namely **Deepseek-LLaMA-8B** and **Phi-4-Reasoning**. Across these models, Parallel-Probe consistently reduces both **total tokens** and **sequential tokens** while maintaining comparable, and in some cases slightly improved, accuracy. These results suggest that the benefits of Parallel-Probe are **not specific to the Qwen family**.
>
> | **Method** | **Total Tokens** | **Sequential Tokens** | **Acc.** |
> | --- | --- | --- | --- |
> | **Deepseek-LLama-8B on AIME25** |  |  |  |
> | SC@64 | 919.5k | 31.4k | 53.3 |
> | Parallel-Probe | 742.5k (-19.2%) | 21.8k (-30.6%) | 53.3 |
> | **Deepseek-LLama-8B on AIME24** |  |  |  |
> | SC@64 | 894.7k | 34.5k | 79.2 |
> | Parallel-Probe | 662.8k (-25.9%) | 18.7k (-45.8%) | 80.7 |
> | **Deepseek-LLama-8B on HMMT25** |  |  |  |
> | SC@64 | 985.7k | 36.7k | 26.7 |
> | Parallel-Probe | 568.3k (-42.3%) | 21.1k (-42.5%) | 27.1 |
> | **Phi-4 Reasoning on AIME25** |  |  |  |
> | SC@64 | 734.6k | 24.7k | 76.7 |
> | Parallel-Probe | 524.7k (-28.6%) | 17.6k (-28.7%) | 76.7 |
>
> **3. Regarding Minor typo / presentation issues**
>
> Thank you for catching this. We will correct the typo from “SC + ASC” to “SC + SAC”. We also agree that including ASC and ESC would improve the clarity of the scaling comparison. We will add the ASC and ESC scaling curves in the final version of the paper for a more complete visual comparison.
>
> **We hope the above clarifications and additional analyses fully address the reviewer’s concerns.**

---

> > ### Author Rebuttal · Reviewer_oM9s · 2026-04-01
> >
> > All of my concerns are fully resolved, and I have raised my score to 5.

---

### Official Review · Reviewer_Ja13 · 2026-03-09

**Soundness:** 3
**Presentation:** 3
**Significance:** 3
**Originality:** 3
**Overall Recommendation:** 4
**Confidence:** 3

**Summary:**

This paper introduces a new way to make AI models think faster and more efficiently by looking at all their parallel reasoning paths at once, rather than just one at a time. The researchers created a tool called "2D probing" that periodically checks the progress of multiple reasoning "branches" to see if they are reaching a common answer.

**Compliance With Llm Reviewing Policy:**

Affirmed.

**Key Questions For Authors:**

how does the system handle cases where the AI is performing very creative tasks where there isn't a single "correct" answer to vote on? The paper focuses heavily on math and logic problems where consensus is easy to measure, but it's unclear if pruning branches would hurt the quality of more open-ended writing or brainstorming. Another question is about the "warmup stage" mentioned; how much does the success of the system depend on picking the right amount of time to wait before starting to prune branches? If the system starts pruning too early, could it accidentally kill off a "slow but genius" reasoning path that would have eventually found the right answer?

**Limitations:**

the "2D probing" interface requires specific intervals to check for answers, which adds its own small overhead to the process. While it saves tokens overall, the actual wall-clock time saved might vary depending on how a specific computer system handles these frequent interruptions. There is also the "onset ratio" mentioned, which shows that while consensus happens early on average, it doesn't happen early for every single problem.

**Strengths And Weaknesses:**

The paper’s primary strength lies in its shift from looking at individual reasoning paths to analyzing the collective behavior of an entire group of parallel branches. By introducing the 2D probing interface, the authors provide a practical way to visualize and exploit the "width-depth" trade-offs that have previously been ignored in efficiency research. The discovery that a global consensus often stabilizes long before the individual models finish their internal monologues is a significant observation that could change how we deploy large-scale reasoning models. The Parallel-Probe framework is particularly impressive because it is "plug-and-play," meaning it does not require expensive retraining or fine-tuning of existing models. This makes the research highly relevant for immediate industry application where reducing the cost of high-quality reasoning is a top priority. Furthermore, the development of the SCOUT simulation framework provides the research community with a valuable tool to test different efficiency strategies without the massive computational overhead usually required for such experiments.

the paper has several weaknesses regarding the consistency and breadth of its findings across different types of AI tasks. A major concern is the "early elicitation" technique, which essentially forces a model to stop its current thought process and provide a final answer prematurely. This may lead to "hallucinations" or incorrect answers on more complex problems where the model actually needs those extra steps to reach a logical conclusion. While the paper shows success on mathematical and logical benchmarks, it lacks evidence that this pruning strategy works for tasks that require deep creative nuance or where there is no single "right" answer. Additionally, the study focuses mostly on models within the Qwen family, which leaves a gap in understanding how these "global dynamics" apply to other architectures like GPT or Claude. There is also a risk that the "majority voting" mechanism used to determine consensus is too simplistic; it treats all branches as equal, even though some branches might be following a much more rigorous and accurate logical path than others.

---

> ### Author Rebuttal · Authors · 2026-03-31
>
> We thank the reviewer for the constructive feedback. We are encouraged that the reviewer finds our work novel, practically useful, and technically solid. Below we address the main concerns point by point.
>
> **1. Regarding Early Elicitation, Premature Pruning, and the Role of Warmup (W1 & Q2)**
>
> **1.1 Warmup mitigates unreliable answers under early elicitation.**
>
> We explicitly anticipated the risk of unreliable answers caused by early elicitation. To this end, we introduce a warmup stage, so that probing is triggered only after sufficient reasoning context has accumulated. This improves the reliability of the probed answers and the resulting consensus signal.
>
> **1.2 Sensitivity to warmup length.**
>
> While we already analyzed warmup in Table 2 and Figure 4, we now provide a more extensive sensitivity study (see table below). We observe that once the warmup length exceeds a moderate threshold (e.g., 15 steps), the performance of Parallel-Probe remains broadly comparable to SC@64 in the vast majority of cases, and in some settings even improves. This suggests that warmup is an important but stable design parameter in practice.
>
> **1.3 Slow-but-correct branches.**
>
> The risk of suppressing a slow-but-correct branch cannot be completely ruled out, but our results suggest that this issue is substantially mitigated once warmup reaches a moderate range, e.g., more than 15 steps. Empirically, Parallel-Probe maintains, and in some cases exceeds, the accuracy of the full-reasoning baseline, suggesting that such branches are not systematically suppressed in practice.
>
> |  | **AIME-24 Qwen3 0.6B** | **AIME-25 Qwen3 0.6B** | **HMMT25 Qwen3 0.6B** | **AIME-24 Qwen3 4B** | **AIME-25 Qwen3 4B** | **HMMT25 Qwen3 4B** | **AIME-24 Qwen3 8B** | **AIME-25 Qwen3 8B** | **HMMT25 Qwen3 8B** |
> | --- | --- | --- | --- | --- | --- | --- | --- | --- | --- |
> | SC@64 | 21.4 | 28.9 | 18.1 | 80.0 | 76.6 | 43.6 | 80.4 | 76.7 | 48.9 |
> | Parallel-Probe warm=5 | 20.2 | **28.9** | 17.1 | 76.0 | 75.5 | 42.4 | **80.6** | 75.5 | 45.0 |
> | Parallel-Probe warm=10 | **21.4** | **29.2** | **18.2** | 76.7 | 75.9 | 43.3 | **81.0** | 75.0 | 46.7 |
> | Parallel-Probe warm=15 | **21.8** | **29.7** | **18.5** | 79.7 | 76.1 | **44.7** | **81.5** | **76.9** | 47.1 |
> | Parallel-Probe warm=20 | **22.3** | **29.9** | **18.5** | **80.2** | **76.9** | **43.7** | **82.5** | **77.1** | 47.6 |
>
> **2. Regarding Generalization Beyond Verifiable Tasks (W2 & Q1)**
>
> We agree that evaluating generalization on creative/open-ended tasks is a valuable direction. Our current work prioritizes well-grounded reasoning tasks with fixed-form answers, as these allow for rigorous, quantitative evaluation via standard majority voting.
>
> However, we clarify our Parallel-Probe is conceptually compatible with open-ended generation. Our improvement to self-consistency **does not depend on** how we combine the final answers. We used "exact-match" for math benchmarks because they have a single clear answer. For open-ended tasks, instead of matching exact words, we can just group the responses by their **meaning [1,2]**. For example, we could group similar answers together or use an "LLM-as-a-judge" to find the consensus before applying our method and use this signal to guide early-stopping and pruning. While implementing this pipeline requires a distinct evaluation setup, we consider it a promising direction for future work.
>
> [1] **Universal Self-Consistency for Large Language Model Generation**
>
> [2]**Latent Self-Consistency for Reliable Majority-Set Selection in Short- and Long-Answer Reasoning**
>
> **3. Regarding Generalization beyond Qwen family (W3)**
>
> To address this concern, we added results on LLaMA-based and Phi-based models, where Parallel-Probe retains similar token-efficiency gains with comparable accuracy; detailed results are provided in Section 2 of our response to Reviewer oM9s. We also observed similar qualitative dynamics trends, suggesting that the global dynamics are not specific to the Qwen family, which we will include in the revised version.
>
> **4. Regarding simplicity of majority voting to determine consensus (W4)**
>
> We agree that more sophisticated aggregation schemes may further improve Parallel-Probe. However, they also introduce additional complexity, overhead, and dependence on auxiliary modules. That said, we believe quality-aware aggregation is a promising direction for future work.
>
> **5. Regarding practical efficiency and onset ratio (Limitation Section)**
>
> **We additionally evaluated Parallel-Probe in an online setting and found that it reduces wall-clock latency in practice; please refer to Section 1.3 of our response to Reviewer oM9s for detailed results.**  Consistent with this, although consensus emerges late for some questions, it tends to form relatively early overall, with a mean onset ratio of 0.31, leaving substantial room for efficiency improvement.
>
> **We hope the above clarifications and additional analyses fully address the reviewer’s concerns.**

---

> > ### Author Rebuttal · Reviewer_Ja13 · 2026-03-31
> >
> > I maintain my positive score

---

### Official Review · Reviewer_xbNR · 2026-03-12

**Soundness:** 3
**Presentation:** 3
**Significance:** 2
**Originality:** 3
**Overall Recommendation:** 4
**Confidence:** 4

**Summary:**

The paper proposes a simple addition to parallel test-time scaling strategies like majority voting by more intelligently allocating token budget to different trajectories within the parallel group. Specifically, the authors propose introducing a probe which forcefully terminates thinking of trajectories using </think> to extract the model's best guess at an answer before natural termination. These extracted answers are used for consensus-based early-stopping when there is mass agreement, and to trim away trajectories that significantly deviate away from the consensus. Experiments demonstrate improved token efficiency compared to the baseline parallel scaling methods without sacrificing accuracy.

**Compliance With Llm Reviewing Policy:**

Affirmed.

**Final Justification:**

The rebuttal has addressed my concerns through additional experiments.

**Key Questions For Authors:**

1. Could you run more evaluations on reasoning-gym tasks (or other non-math reasoning benchmarks)
2. Forcefully terminating thinking with </think> will not work for instruct models like qwen3-4b-instruct which do not reason within thinking tags. Do the authors have ideas about hoow to force answer extraction in these cases? Could some experiments be done with qwen3-4b-instruct?

If the authors can address these questions I can raise the score to weak accept.

**Limitations:**

The authors do not explicitly discuss limitations of their work.

**Strengths And Weaknesses:**

Strengths:
1. The method is simple and well-motivated. I can see it practically being used with majority voting to save token costs
2. The baselines are satisfactory and token savings demonstrated in the experiments are solid.

Weaknesses:
1. The primary weakness in my opinion is limiting the experiments to AIME/HMMT math datasets. Considering this is a purely empirical paper studying an improved test-time scaling strategy, I would like to see more comprehensive evals on reasoning tasks of other types. My suggestion to the authors is to also evaluate on some reasoning-gym problems (_Stojanovski et al._)
2. The method is limited to problem types where answers can be easily compared/clustered, similar to majority voting. This does not extend to tasks where independent answers cannot be directly compared (eg. code generation, proofs).
3. The methods is fairly simple and while I think it is useful, it is a minor contribution all things considered.

---

> ### Author Rebuttal · Authors · 2026-03-31
>
> We thank the reviewer for the constructive feedback. We are encouraged that the reviewer finds the method simple, well-motivated, and practically useful. Below we address the main concerns point by point.
>
> **1. Regarding Broader evaluation beyond AIME/HMMT, e.g., reasoning-gym or other non-math reasoning tasks. (W1 and Q1)**
>
> To address the concern about generalization beyond math olympiad benchmarks, we further add experiments on **GPQA-Diamond**, a challenging **non-math reasoning benchmark** and **Reasoning Gym**.
>
> **Setups for Reasoning Gym:** we generate a total of **30 problems** across **three task categories**: **maze**, **rotten_oranges**, and **zebra_puzzles**, with **10 examples per category**.
>
> The results for **Qwen-3-0.6B** are shown below. Across these non-math reasoning tasks, Parallel-Probe shows a similar pattern of **reducing both total and sequential tokens while preserving comparable accuracy**, suggesting that the method generalizes beyond math olympiad benchmarks.
>
> | Method | Type | Acc. ↑ | SeqTokens ↓ | Total Tokens  ↓ |
> | --- | --- | --- | --- | --- |
> | Qwen-3-0.6B on Reasoning-Gym |  |  |  |  |
> | SC @ 64 | Parallel | 57.2 | 17.2k | 320.1k |
> | ASC | Seq. | 57.4 | 153.5k (+793.6%) | 153.5k (-52.1%) |
> | ESC | Hybrid | 57.3 | 62.9k (+266.2%) | 252.4k (-21.1%) |
> | SC @ 64 + SAC | Parallel | 57.1 | 11.2k (-34.6%) | 290.7k (-9.2%) |
> | Parallel-Probe | Parallel | 57.2 | 6.5k (-62.2%) | 267.7k (-16.4%) |
>
> | Method | Type | Acc. ↑ | SeqTokens ↓ | Total  Tokens ↓ |
> | --- | --- | --- | --- | --- |
> | Qwen-3-0.6B on GPQA-Diamond |  |  |  |  |
> | SC @ 64 | Parallel | 28.7 | 15.4k | 339.4k |
> | ASC | Seq. | 28.7 | 157.3k (+919.7%) | 157.3k (-53.7%) |
> | ESC | Hybrid | 28.7 | 63.8k (+313.5%) | 302.2k (-11.0%) |
> | SC @ 64 + SAC | Parallel | 26.4 | 10.7k (-30.9%) | 317.3k (-6.5%) |
> | Parallel-Probe | Parallel | 30.1 | 7.6k (-50.7%) | 279.6k (-17.6%) |
>
> **2. Regarding applicability to instruct models without thinking tags (e.g., Qwen3-4B-Instruct). (Q2)**
>
> **Parallel-Probe does not rely on thinking tags itself**. Its core function is to extract a **quick answer estimate from a partial trajectory** for consensus checking and pruning. For instruct models, this can be done by **prompting the model to directly provide its best current final answer based on the partial reasoning so far**, rather than by forcibly closing a reasoning block.
>
> To address the reviewer’s concern, we also added experiments on **Qwen3-4B-Instruct** and AIME25. **The results provide initial evidence that Parallel-Probe can also be applied to instruct models, and is not restricted to models with explicit thinking tags.**
>
> | **Method** | **Acc.** | **SeqTokens** | **Tokens** |
> | --- | --- | --- | --- |
> | SC@64 | 65.0 | 25.8k | 509.4k |
> | Parallel-Probe | 65.0 | 11.1k (-57.0%) | 408.4k (-19.8%) |
>
> **3. Regarding applicability to problems where independent answers cannot be directly compared (eg. code generation, proofs). (W2)**
>
> Thank you for pointing this out. We agree that evaluating generalization on open-ended tasks is a valuable direction. Our current work prioritizes well-grounded reasoning tasks with fixed-form answers, as these allow for rigorous, quantitative evaluation via standard majority voting.
>
> However, we clarify our Parallel-Probe is conceptually compatible with open-ended generation. Our improvement to self-consistency **does not depend on** how we combine the final answers. We used "exact-match" for math benchmarks because they have a single clear answer. For open-ended tasks, instead of matching exact words, we can just group the responses by their **meaning [1,2]**. For example, we could group similar answers together or use an "LLM-as-a-judge" to find the consensus before applying our method and use this signal to guide early-stopping and pruning. We consider it a promising direction for future work and will include a discussion in the Limitations and Future Work section of the next version of the paper.
>
> [1] **Universal Self-Consistency for Large Language Model Generation**
>
> [2]**Latent Self-Consistency for Reliable Majority-Set Selection in Short- and Long-Answer Reasoning**
>
> **4. On the contribution size. (W3)**
>
> Thank you for acknowledging the lightweight and practical nature of Parallel-Probe. That said, we believe this work makes an important contribution to efficient reasoning and parallel thinking. Rather than treating each branch as an independently sampled trajectory whose only meaningful signal appears at the end, we reveal that informative intermediate signals and cross-branch dependencies emerge during decoding, and empirically show that exploiting them leads to improved reasoning efficiency.
>
> **We hope the above clarifications and additional analyses fully address the reviewer’s concerns.**

---

> > ### Author Rebuttal · Reviewer_xbNR · 2026-04-03
> >
> > I have increased the score appropriately.

---

### Official Review · Reviewer_EJeP · 2026-03-16

**Soundness:** 3
**Presentation:** 3
**Significance:** 2
**Originality:** 3
**Overall Recommendation:** 3
**Confidence:** 2

**Summary:**

The authors introduce 2D Probing/SCOUT/parallel-probe, methods for efficient parallel thinking that injects a EOS token during the CoT, which can be used to prune redundant steps. The authors demonstrate that parallel-probe reduces sequential tokens more than 30% and total token cost over 20% compared to self-consistency.

**Compliance With Llm Reviewing Policy:**

Affirmed.

**Key Questions For Authors:**

see weaknesses above

**Limitations:**

yes

**Strengths And Weaknesses:**

* The finding from observation 2 is very surprising, in that the total computation cost of often dominated by a few outlying trajectories. This serves as an excellent motivation for the method that the authors propose.
* SCOUT is a reasonable testbed for future works on efficient inference-time algorithms to be tested on.

As for weaknesses,
* The experiments are conducted on math olympiad benchmarks only, namely, AIME 2024/2025, HMMT 2025. It is questionable if the method could be applied on non-verifiable tasks where there is no decisive answer (e.g., a numerical answer) but rather an open-ended response.
* Also, the experiments are conducted only with Qwen3 model familes, which raises concerns if the proposed method also works for Llama, GPT-OSS, Kimi-K2, and other models.
* In addition to total tokens, it would be nice if there is FLOPs or time for each of the baseline methods.

---

> ### Author Rebuttal · Authors · 2026-03-31
>
> Thank you for taking time reviewing our paper and providing insightful feedback. We are glad to address your concerns point by point.
>
> **1. Regarding Generalization beyond Qwen family**
>
> To directly address this concern, we evaluated Parallel-Probe on two non-Qwen model families, including DeepSeek-LLaMA-8B and Phi-4-Reasoning. We run DeepSeek-LLaMA-8B on all three tasks, and additionally evaluate Phi-4-Reasoning on AIME25.
>
> **As shown in the table below, Parallel-Probe generalizes well beyond** **Qwen family:** Parallel-Probe consistently reduces both total and sequential tokens while preserving comparable accuracy.
>
> | **Method** | **Total Tokens** | **Sequential Tokens** | **Acc.** |
> | --- | --- | --- | --- |
> | **Deepseek-LLama-8B on AIME25** |  |  |  |
> | SC@64 | 919.5k | 31.4k | 53.3 |
> | Parallel-Probe | 742.5k (-19.2%) | 21.8k (-30.6%) | 53.3 |
> | **Deepseek-LLama-8B on AIME24** |  |  |  |
> | SC@64 | 894.7k | 34.5k | 79.2 |
> | Parallel-Probe | 662.8k (-25.9%) | 18.7k (-45.8%) | 80.7 |
> | **Deepseek-LLama-8B on HMMT25** |  |  |  |
> | SC@64 | 985.7k | 36.7k | 26.7 |
> | Parallel-Probe | 568.3k (-42.3%) | 21.1k (-42.5%) | 27.1 |
> | **Phi-4 Reasoning on AIME25** |  |  |  |
> | SC@64 | 734.6k | 24.7k | 76.7 |
> | Parallel-Probe | 524.7k (-29.8%) | 17.6k (-28.7%) | 76.7 |
>
> **2. Regarding more evaluation metrics, e.g., time or FLOPs**
>
> To directly address this concern, we have added online evaluation and reported the average wall-clock time per question for Parallel-Probe, SC@64, and SC@64+SAC under the same hardware and serving setup (4 x H200 GPUs).
>
> We show the results on AIME25 below.
>
> **Findings**:
>
> - **Parallel-Probe is practically efficient.** In our experiments, it achieves **lower average wall-clock time** than the **SC@64** baseline while maintaining comparable performance.
> - **SAC + SC@64 is also faster than SC@64, but remains slower than Parallel-Probe.** This suggests that leveraging cross-branch signals enables more effective early stopping and pruning.
>
> | **Method** | **Wall-clock time (s/problem)** | **AIME25 Acc.** |
> | --- | --- | --- |
> | **Qwen-3-0.6B** |  |  |
> | SC@64 | 170.0 | 28.3 |
> | SAC + SC@64 | 163.4 | 25.0 |
> | Parallel-Probe | **113.2** | 30.0 |
> | **Qwen-3-4B** |  |  |
> | SC@64 | 260.2 | 76.7 |
> | SAC + SC@64 | 222.6 | 70.0 |
> | Parallel-Probe | **155.8** | 76.7 |
> | **Qwen-3-8B** |  |  |
> | SC@64 | 273.1 | 76.7 |
> | SAC + SC@64 | 227.9 | 70.0 |
> | Parallel-Probe | **196.9** | 76.7 |
>
> **3. Regarding applicability of Parallel-Probe in non-verifiable tasks where there is no decisive answer (e.g., a numerical answer) but rather an open-ended response).**
>
> Thank you for pointing this out. We agree that evaluating generalization on open-ended tasks is a valuable direction. Our current work prioritizes well-grounded reasoning tasks with fixed-form answers, as these allow for rigorous, quantitative evaluation via standard majority voting.
>
> However, we clarify our Parallel-Probe is conceptually compatible with open-ended generation. Our improvement to self-consistency **does not depend on** how we combine the final answers. We used "exact-match" for math benchmarks because they have a single clear answer. For open-ended tasks, instead of matching exact words, we can just group the responses by their **meaning [1,2]**. For example, we could group similar answers together or use an "LLM-as-a-judge" to find the consensus before applying our method and use this signal to guide early-stopping and pruning. While implementing this pipeline requires a distinct evaluation setup, we consider it a promising direction for future work.
>
> [1] **Universal Self-Consistency for Large Language Model Generation**
>
> [2] **Latent Self-Consistency for Reliable Majority-Set Selection in Short- and Long-Answer Reasoning**
>
> **We hope the above clarifications and additional analyses fully address the reviewer’s concerns.**

---

### Decision · Program_Chairs · 2026-04-30

**Decision:**

Accept (regular)

**Comment:**

This paper proposes a simple and practically useful method for improving the efficiency of parallel test-time scaling by probing intermediate branch states, detecting emerging consensus, and pruning redundant or off-consensus trajectories before full completion. Reviewers found the core idea well motivated, the 2D probing perspective insightful, and the SCOUT framework a useful contribution for studying efficient inference-time strategies. The main initial concerns were about breadth of evaluation, latency reporting, and generalization beyond Qwen/math-only settings, but the rebuttal addressed these with additional experiments on non-Qwen models, non-math benchmarks, instruct models, and online wall-clock measurements. Overall, the paper makes a solid empirical contribution with clear practical value, and I recommend accept.